Warming resistant corals from the Gulf of Aqaba live close to their cold-water bleaching threshold

Bellworthy Jessica jhbellworthy@gmail.com
Fine Maoz
1 The Goodman Faculty of Life Sciences, Bar Ilan University , Ramat Gan , Israel
2 The Interuniversity Institute for Marine Sciences in Eilat , Eilat , Israel
Ries Leslie
Electronic publication date: 2021 Mar 25
Publication date: 2021
Volume: 9
Electronic Location ID: e11100
Received 2020 Nov 16; Accepted 2021 Feb 22
Copyright: ©2021 Bellworthy and Fine
Copyright year: 2021
Copyright holder: Bellworthy and Fine
License: This is an open access article distributed under the terms of the Creative Commons Attribution License, which permits unrestricted use, distribution, reproduction and adaptation in any medium and for any purpose provided that it is properly attributed. For attribution, the original author(s), title, publication source (PeerJ) and either DOI or URL of the article must be cited.
License URL: https://creativecommons.org/licenses/by/4.0/

Keywords: Acropora eurystoma, Coral physiology, Photosynthesis, Red Sea, Stylophora pistillata, Symbiont

Funding: An Israel Science Foundation grant #1794/16 Binational USA-Israel Science Foundation #2016403 This study was supported by an Israel Science Foundation grant (#1794/16) and Binational USA-Israel Science Foundation (#2016403) to Maoz Fine. There was no additional external funding received for this study. The funders had no role in study design, data collection and analysis, decision to publish, or preparation of the manuscript.

==============================
Global climate change is causing increasing variability and extremes in weather worldwide, a trend set to continue. In recent decades both anomalously warm and cold seawater temperatures have resulted in mass coral bleaching events. Whilst corals’ response to elevated temperature has justifiably attracted substantial research interest, coral physiology under cold water stress is relatively unfamiliar. The response to below typical winter water temperature was tested for two common reef building species from the Gulf of Aqaba in an ex situ experiment. Stylophora pistillata and Acropora eurystoma were exposed to 1 or 3 °C below average winter temperature and a suite of physiological parameters were assessed. At 3 °C below winter minima (ca. 18.6 °C), both species had significant declines in photosynthetic indices (maximum quantum yield, electron transport rate, saturation irradiance, and photochemical efficiency) and chlorophyll concentration compared to corals at ambient winter temperatures. It was previously unknown that corals at this site live close to their cold-water bleaching threshold and may be vulnerable as climate variability increases in magnitude. In order to determine if a cold winter reduces the known heat resistance of this population, the corals were subsequently exposed to an acute warm period at 30 °C the following summer. Exposed to above typical summer temperatures, both species showed fewer physiological deviations compared to the cold-water stress. Therefore, the cold winter experience did not increase corals’ susceptibility to above ambient summer temperatures. This study provides further support for the selection of heat tolerant genotypes colonising the Red Sea basin and thereby support the mechanism behind the Reef Refuge Hypothesis.

Introduction

Coral reefs form within defined environmental boundaries (Kleypas, Mcmanusi & Menez, 1999). Reef building corals are mostly limited to the tropics as low winter water temperature limits the expansion of coral reefs to higher latitudes (Higuchi et al., 2015; Nakabayashi et al., 2019). Dana (1843) first suggested 19 °C to be the lower temperature limiting the distribution of coral reefs. Later field and lab observations lowered this absolute value to 18 °C (Vaughan & Wells, 1943), 16 °C (Kleypas, Mcmanusi & Menez, 1999), or 14 °C (Mayor, 1914).

Ocean warming is considered the main cause of recurrent mass coral bleaching events and the recent decline of coral reefs worldwide (Hoegh-Guldberg & Poloczanska, 2017). Coral bleaching is a potentially fatal reduction in the number of algal symbionts and/or the photosynthetic pigment chlorophyll following anomalous water temperature or light levels. Cold water bleaching is much less frequently reported (but see e.g., Hoegh-Guldberg & Fine, 2004; Lirman et al., 2011; Kemp et al., 2011; Zapata, Jaramillo-González & Navas-Camacho, 2011). Fewer studies have been conducted on the effect of cold-water stress in corals although global anthropogenic climate change has and will continue to produce more extreme weather events with increased variability (Urban, Cole & Overpeck, 2000), including cold episodes (Johnson et al., 2018). Low temperatures can result in significant changes in coral physiology, bleaching, and mortality (Saxby, Dennison & Hoegh-Guldberg, 2003; Kemp et al., 2011; Roth, Goericke & Deheyn, 2012). Corals’ response to cold stress, as is the case for heat stress, appears to vary regionally likely as a consequence of local adaptation, but also as an intricate combined function of the magnitude and duration of stress. In 1977, a cold-air front reduced water temperatures in Florida Bay to below 16 °C for 8 consecutive days causing mortality of up to 91% of corals at some reef sites (Roberts et al., 1982). Another prolonged cold-water period with water temperatures below 11 °C in 2010 resulted in 26% mean coral mortality in the Upper Florida Keys (Lirman et al., 2011; Kemp et al., 2016). Three consecutive months of below average winter minima (ca. 19 °C) caused approximately 60% of corals to bleach in Loreto in the Sea of Cortez in 2006 (LaJeunesse, Reyes-Bonilla & Warner, 2007). A similar event in 2008 with temperatures below 17 °C caused bleaching of up to 90% of Pocilloporid corals in the same region (LaJeunesse et al., 2010). In regions where corals are at risk from both hot and cold bleaching, such as the southern Great Barrier Reef (Hoegh-Guldberg & Fine, 2004; Hoegh-Guldberg et al., 2005), there exists the possibility that reefs could experience two bleaching events within the same year, dramatically reducing vital recovery windows between thermal stressors. Furthermore, studies show that if cold stress occurs in combination with other global environmental changes, such as ocean acidification, the risk of coral mortality increases compared to low temperature stress in isolation (Kavousi, Parkinson & Nakamura, 2016).

The Gulf of Aqaba (GoA) and northern Red Sea is a region highlighted for its lack of bleaching both during recent anomalously warm years in the field and during ex situ experiments (Bellworthy & Fine, 2017; Krueger et al., 2017; Osman et al., 2018). This coral population is postulated to be adapted to warmer than present day seawater temperatures due to a warm water selection barrier at the entrance to the Red Sea, the Bab El Mendeb in the south (Fine, Gildor & Genin, 2013). Experiments using GoA coral species have shown increased net oxygen production (proxy for algal symbiont productivity) at above ambient summer temperatures (Krueger et al., 2017). However, the response of these corals to cold stress events has not been tested. The reefs of the GoA are some of the northernmost in the world; limited research has suggested that high latitude corals may have greater tolerance to cold stress than their tropical conspecifics (Tuckett & Wernberg, 2018). In the northernmost part of the GoA, a steep slope from the forereef to a depth of 40-100 m may prevent major cold-water events on the shallow reef since denser cooler water will sink to greater depth. Yet, under doldrums conditions in lagoons and reef-flats, extreme cold conditions may develop. Furthermore, climate models for this region suggest that the Red Sea will soon enter a cooling phase (Krokos et al., 2019).

Since bleaching severity following cold water stress has previously been shown to be species specific (Coles & Fadlallah, 1991; Zapata, Jaramillo-González & Navas-Camacho, 2011; Kemp et al., 2016; Pontasch et al., 2017; Tuckett & Wernberg, 2018), the responses of two species were compared in this study. Acroporid corals are the most abundant scleractinian genera in Eilat with Stylophora sp. the second most abundant (Shaked & Genin, 2018). Acropora eurystoma is endemic to the Red Sea. Stylophora pistillata is particularly well studied at this site. Importantly, the response to heat stress of both species has been previously characterised (Bellworthy & Fine, 2017). Both species are key reef building species, and their branching structure provides important habitat for many reef fish and invertebrates. The study was designed to answer two questions: 1. How do two species from the GoA respond to below winter minimum water temperatures? 2. Does a recent cold winter experience make these corals more vulnerable to an acute warm period during summer?

Materials & Methods

Experimental design

Corals were collected from the coral nursery (8 –12 m depth) adjacent to the Interuniversity Institute (IUI) for Marine Sciences in Eilat, Israel (Gulf of Aqaba, Red Sea, 29°30′N, 34°55′E). Sampling was conducted under permit no. 2018/41868 from the Israel Nature Parks Authority. Average surface seawater temperature in the month prior to coral collection (February 2018) was 21.70 ± 0.09 °C (Israel National Monitoring Program of the Gulf of Eilat, 2020). The minimum monthly mean temperature between 2011 –2018 for the collection site was 21.41 ± 0.36 °C (mean ± standard deviation, data available: http://www.meteo-tech.co.il/eilat-yam/eilat_daily_en.asp). Sea surface temperatures at the collection site do not often fall below 21 °C (Genin, Lazar & Brenner, 1995; Israel National Monitoring Program of the Gulf of Eilat, 2020); the minimum single point temperature since records began in 2007 (10 min sampling interval) is 20.4 °C. The month with the coldest average temperature was February 2008 (20.64 ± 0.12 °C; mean ± standard deviation).

Two species, Acropora eurystoma Klunzinger 1879 and Stylophora pistillata Esper 1797, were collected. At this collection depth, S, pistillata likely hosted Symbiodiniaceae symbionts from the genus Symbiodinium, formally clade A (Byler et al., 2013) and A. eurystoma with Cladocopium sp., formly clade C (Karako-Lampert et al., 2004; LaJeunesse et al., 2018). Six genotypes of each species with healthy appearance (i.e., no signs of bleaching, algal growth, or predation) were selected. From each colony (n = 6 colonies /species) 10 fragments of ca. three cm length were sampled resulting in a total of 60 fragments per species. Coral fragments were brought to the Red Sea Simulator experimental system (Bellworthy & Fine, 2018) and placed in flow through aquaria with water sourced from the collection site. Twenty-four hours after collection, corals were glued to bases to hold fragments upright and to facilitate handling.

Fragments were assigned to adjacent experimental aquaria (n = 9 aquariums). One fragment from each genotype was placed randomly in each of the nine aquaria. This resulted in a total of 12 fragments in each 40 L aquarium (6 fragments from each of two species). Fragments remained in ambient seawater for two weeks acclimation to tank conditions, prior to the onset of cooling. During this ‘acclimation’ period the corals were exposed to 290 µmol m−2 s−1 photosynthetically active radiation (PAR) at midday which approximates the midday PAR the collection site in winter (Dishon et al., 2012). Aquarium water temperatures for all phases of the experiment are shown in Table 1.

Table 1 Automated temperature recordings in experimental aquaria ca. every 5 minutes throughout the experiment (every 10 minutes during ambient recovery phase).

Temperature change refers to the rate of temperature decrease from ambient in the cold stress ’winter’ phase. Values are treatment averages of all recorded values (day and night) for each period from triplicate aquaria combined (mean ± standard deviation). Acclimation refers to the days corals were in experimental aquaria before onset of cooling. Winter refers to the 16-day cold stress experimental period at minimal temperatures (i.e. 28th March–12th April). Recovery: two months in ’common garden’ aquaria at ambient seawater temperature. Summer refers to acute heat shock (maximal temperature hold days) 11th–13th June. During the summer phase, fragments from all winter treatments were combined into three replicate heated aquaria with a common above ambient temperature.

Treatment	Temperature change (°C d−1)	Temperature (°C)	
		Acclimation	Winter	Recovery	Summer	
T0	−0.0	20.27 ± 0.37	22.42 ± 0.19	24.74 ± 1.28	29.93 ± 0.93	
T-1	−0.3	20.70 ± 0.65	20.53 ± 0.46			
T-3	−0.6	20.38 ± 0.45	18.61 ± 0.77			

In order to assess how these two species from the GoA respond physiologically to cold water below ambient winter temperatures, a cold stress experiment was initiated in late-March. Corals were exposed to three discrete treatment temperatures in a ’winter’ phase (Table 1): ambient control (T0), 1 °C below winter minima (T-1), and 3 °C below winter minima (T-3). All experimentation was conducted using the Red Sea Simulator (Bellworthy & Fine, 2018) where temperature was independently controlled in each aquarium. User defined settings and environmental data were viewed and logged through custom software (Crystal OPC, CrystalVision, Samar, Israel). Each treatment was replicated in three (triplicate) aquaria. Ambient control aquaria (T0) were supplied with filtered seawater (500µm), direct from the collection site, without temperature manipulation (22.42 ± 0.19 °C). In T-1 and T-3 aquaria, water temperature was decreased at 0.3 and 0.6 °C/ day, respectively, for 7 days. The cooling rate is substantially more moderate than cold stress events reported in the field (Walker et al., 1982; Lirman et al., 2011) and compared to previous empirical tests (Roth, Goericke & Deheyn, 2012; Pontasch et al., 2017) which reduce water temperature by up to 7 °C in 11 h. It is therefore assumed that the cooling rate itself did not cause meaningful stress.

Following cooling, experimental temperatures of 20.53 ± 0.46 °C and 18.61 ± 0.77 °C (ca. 1.1 °C (T-1) and 3.2 °C (T-3) below average winter temperature in 2018, respectively) were maintained for 16 days. Cooling rate and experimental duration was chosen to be directly comparable (but opposite) to a previous heat stress experiment at this site (Bellworthy & Fine, 2017) but also relevant to the duration of cold-water events that have caused bleaching in the field e.g., Florida Keys, January 2010 (Lirman et al., 2011). Desired temperature was set at an offset from incoming seawater in order to maintain diel variability in aquaria.

After 16-days cold stress, seawater temperature was returned to ambient for all remaining corals at the same rate as cooling occurred. Corals from all treatments were then maintained in ‘common garden’ aquaria with ambient seawater conditions for two months (13th April–11th June 2018). Average seawater temperature during this period was 24.74 ± 1.28 °C and increased from ca. 22.8 to 25.7 °C with natural seasonal warming.

The second experimental phase (mid-June) sought to investigate whether corals’ cold winter experience had any impact on response to a subsequent abrupt and acute warming event. In this ’summer’ phase, remaining fragments (N total = 36; one fragment per genotype (n = 6 genotypes), per species (n = 2), per winter stress treatment (n = 3)) were exposed, in triplicate aquaria, to above local maximum monthly mean temperature. The summer phase consisted of a single common above ambient temperature for all corals irrespective of their winter treatment (Table 1). This period lasted for 7 days incorporating four days thermal ramping at 1 °C day−1 followed by three days thermal hold.

Sampling and physiological analyses

One coral fragment from each genotype and each species was sampled from each treatment at the end of each experimental period (winter and summer). A range of physiological parameters was assessed on each fragment as previously described (Bellworthy & Fine, 2017). Briefly, corals were dark acclimated for 20 min before a rapid light curve was performed on every fragment (14 saturation pulses under increasing actinic light from 0 –700 µmol m−2 s−1, Imaging PAM, Walz GmbH, Germany). Maximum quantum yield (FV/FM), is theoretically independent from chlorophyll concentration and because samples are dark acclimated, is also independent from ETR (Ralph & Gademann, 2005). The equation of Hill et al. (2004) was used in conjunction with the regression wizard tool of SigmaPlot v12 (SigmaStat, LaJolla, CA, USA) to derive the parameters maximal relative electron transport rate (rETRMAX), saturating irradiance (Ik), and initial slope of rETR under light limited conditions (α; also known as photosynthetic efficiency).

Subsequently, fragments were incubated in sealed, temperature-controlled chambers to determine respiration (oxygen consumption) and net photosynthesis (oxygen production) rates. Respiration was measured in the dark (<2 µmol m−2 s−1) and net photosynthesis at 180 µmol m−2 s−1. Chambers were placed on magnetic stirrers and fitted with a fibre optic oxygen sensor (PreSens) connected to an OXY-4 mini transmitter (PreSens) to follow change in oxygen concentration with incubation time. After incubations, fragments were individually wrapped in aluminium foil and flash frozen before storage at −80 °C. Coral fragments were subsequently processed for protein concentration (Bradford, 1976), chlorophyll a concentration (Jeffrey & Humphrey, 1975), symbiont cell counts (FACS Attune NxT; cell identification by fluorescence and size), and surface area utilising the wax dip method (Stimson & Kinzie, 1991).

Statistical analyses

Raw data were first visualized in order to assess distribution and inspect potential outliers. The parametric assumption of normal distribution was tested for each treatment group within the winter and summer periods separately using the Shapiro–Wilk normality test (shapiro.test, R package: {stats} (R Core Team, 2020)). The assumption of homogenous variances of data between each treatment was tested for each winter and summer period using Levene’s Test (leveneTest, R package: {car} (Fox & Weisberg, 2019)). No extreme violations were present in the raw data.

Linear mixed effects models (lmer, R package: {lme4}(Bates et al., 2015)) were performed for each dependent variable in order to assess differences between treatment groups for each experimental period (i.e., winter and summer), with the fixed factor ‘treatment’ and the random factor ‘genotype’. P-values and F-statistics were obtained using the command ‘anova’. The contributions of the fixed and random factors to the model fit were assessed using the r.squaredGLMM function (R package: {MuMIn}(Bartoń, 2020)). The marginal R2 value indicates the proportion of variance explained by fixed factors alone, whereas the conditional R2 value indicates the proportion of variance explained by both fixed and random factors combined, thereby, in the current case, enabling identification of genotype effects. Model assumptions of normally distributed residuals and equal variance of residuals were tested. When assumptions were met and the model result was significant, post hoc tests were conducted to elucidate which treatment groups were significantly different (emmeans and pwpm, R package: {emmeans}, adj = “tukey” (Lenth, 2020)). Where model assumptions were violated, permutations were performed (permanova.lmer, R package: {predictmeans}) (Luo, Ganesh & Koolaard, 2020). All statistical tests and graphics (R package: {ggplot2} (Wickham, 2016)) were produced in RStudio version 3.6 (https://www.r-project.org/ (R Core Team, 2020)). All raw data and R code for this publication are available at https://github.com/JessicaBellworthy/Bellworthy_Corals_Eilat_ColdStress.

Figure 1 Response of algal symbionts of Stylophora pistillata (A–C) and Acropora eurystoma (D–F) exposed to below annual minima temperatures.

yellow = 22.4 °C, blue = 20.5 °C, grey = 18.6 °C (n = 5 or 6 /species /treatment). Box fill colours reflect the different winter thermal experience of these corals. During ‘summer’ all corals were exposed to 3 °C above maximum monthly mean. Plots are symbiont cell density (A, D), chlorophyll concentration (B, E) and rate of net oxygen production (C, F). Boxes display the median line, the first and third quartiles (box outline) and whiskers are 1.5 times the interquartile range. Shaped points refer to coral genotypes, consistent to all plots. Different lowercase letters denote significant treatment effects within each experimental phase (winter and summer, Linear Mixed Effect Model and Tukey post hoc test).

Results

Temperature manipulation

During the 16-day winter period, all three treatments maintained similar diel variability in temperature. Approximately 2  °C separated mean treatment values (Table 1). As incoming natural reef seawater temperature warmed through the cold stress phase, T-1 represents an extended cold winter, and T-3 represents a sustained below ambient cold stress. During the summer phase, corals from all winter treatments experienced an above summer ambient temperature of 29.93 ± 0.93 °C. During this period the ambient reef seawater temperature was 26.90 ± 0.55 °C (Israel National Monitoring Program of the Gulf of Eilat, 2020) and therefore the experimental temperature was ca. 3 °C above ambient.

Physiological responses to cold stress

In both Stylophora pistillata and Acropora eurystoma cold exposure significantly changed most of the examined physiological parameters compared to corals maintained in ambient seawater. Net photosynthesis in particular was almost non-existent in corals maintained at ∼18 °C (A. eurystoma: 0.003 ± 0.003 µmol hr−1 cm−2; S. pistillata: 0.016 ± 0.020 µmol hr−1 cm−2, Fig. 1). In three and two out of the six genotypes tested, there was no net oxygen production at this temperature for A. eurystoma and S. pistillata respectively (see data points below zero in Figs. 1C and 1F). All photosynthetic parameters derived from light curves were significantly different at ∼18 °C compared to ambient winter temperature (∼22 °C). For example, in A. eurystoma, maximal quantum yield (FV/ FM) was significantly reduced from a mean value of 0.458 ± 0.022 at ambient to 0.359 ± 0.030 at ∼18 °C (post hoc difference: p = 0.004, Fig. 2). In addition, cold stress had a significant impact on chlorophyll concentration in both species e.g., in S. pistillata reducing the ambient concentration of 39.28 ± 6.19 µg cm−2 to 19.91 ± 3.44 µg cm−2 in the coldest treatment (T-3, ∼18 °C, Fig. 1B). Protein concentration and respiration rate were not significantly different between treatments for either species (Fig. 3). For A. eurystoma symbiont cell density was also not significantly impacted by cold stress but for S. pistillata symbiont cell density was significantly reduced at both below ambient temperatures relative to ambient winter temperatures (Fig. 1, Table 2).

Statistical analyses suggest that coral genotype contributed near zero variation to the data for six out of the nine parameters tested following cold stress in S. pistillata and five out of nine parameters in A. eurystoma, i.e., marginal and conditional R2 values were equal (see methods for explanation, Table 2). For A. eurystoma in the winter phase, variables with the greatest R2 values were rETRmax (conditional R2 = 0.846) and Ik (conditional R2 = 0.828). Similarly, for S. pistillata winter variables with the greatest R2 values were Ik (conditional R2 = 0.918) and net photosynthesis rate (conditionalR2 = 0.832), closely followed by rETRmax (conditional R2 = 0.830).

Impacts of cold stress on response to an acute warm period

Compared to the winter phase of the experiment, the summer phase resulted in fewer significant differences between treatments. In other words, in most cases, corals’ summer response was independent of the winter temperature they experienced. The physiological response during the warm period following cold exposure was different in the two species. For A eurystoma at 30 °C, only rETRmax and symbiont cell density were significantly lower in corals that were exposed to 18 °C in the winter compared to those that remained in ambient conditions (Figs. 1 and 2). For S. pistillata at 30 °C, Ik was significantly higher in corals from the 20 °C (T-1) winter treatment and α values were significantly lower in the 20 °C (T-1) and 18 °C (T-3) corals compared to the T0 treatment (Fig. 2). Correspondingly, the variables with the largest conditional R2 were rETRmax (0.676) and photosynthetic α (0.800) for A. eurystoma and S. pistillata respectively. All other parameters were not significantly different between treatments (T0, T-1, T-3) during the summer phase.

Figure 2 Photophysiology of algal symbionts of Stylophora pistillata (A–D) and Acropora eurystoma (D–H) exposed to below annual minima temperatures.

yellow = 22.4 °C, blue = 20.5 °C, grey = 18.6 °C (n = 5 or 6 /species /treatment). Box fill colours reflect the different winter thermal experience of these corals. During ’summer’ all corals were exposed to 3°C above maximum monthly mean. FV∕FM: Maximum quantum yield of photosystem II (A, E). rETRMAX: Maximum rate of electron transport between photosystems (B, F). α: photosynthetic efficiency (C, G). IK: saturating irradiance for photosynthesis, µmol m−2s−1 (D, H). Boxes display the median line, the first and third quartiles (box outline) and whiskers are 1.5 times the interquartile range. Shaped points refer to coral genotypes, consistent to all plots. Different lowercase letters denote significant treatment effects within each experimental phase (winter and summer, Linear Mixed Effect Model and Tukey post hoc test).

Discussion

Stylophora pistillata and Acropora eurystoma were exposed to below ambient seawater temperatures in winter (March), followed by an acute warm period the following summer (June). These corals displayed physiological signs akin to a bleaching response below typical winter temperatures. Photophysiology, chlorophyll concentration, and net photosynthesis (oxygen production) declined (Figs. 1 and 2). Despite this, all corals survived the 16-day cold stress and later showed few physiological differences at 3 °C above ambient compared to corals that did not experience a prior cold stress. These data indicate that experience of cold winter stress does not significantly reduce the known resistance of this population to above ambient summer temperatures. However, this study also shows for the first time that this population may be living critically close to its lower bleaching threshold which appears to represent a sublethal stress, at least in the short term, since host biomass (as measured by protein concentration) was maintained.

Figure 3 Response of Stylophora pistillata (A–B) and Acropora eurystoma (C–D) exposed to below annual minima temperatures.

yellow = 22.4 °C, blue = 20.5 °C, grey = 18.6 °C (n = 5 or 6 /species /treatment). Box fill colours reflect the different winter thermal experience of these corals. During ‘summer’ all corals were exposed to 3°C above maximum monthly mean. Plots are holobiont protein concentration (A, C) and rate of oxygen consumption in the dark i.e., respiration rate (B, D). Boxes display the median line, the first and third quartiles (box outline) and whiskers are 1.5 times the interquartile range. Shaped points refer to coral genotypes, consistent to all plots. Different lowercase letters denote significant treatment effects within each experimental phase (winter and summer, Linear Mixed Effect Model and Tukey post hoc test).

Following cold stress, most significant changes were observed within symbiont (cell density and chlorophyll concentration) and photophysiology parameters rather than coral host physiology (protein concentration and respiration; Table 2). This suggests that, for this population, the algal symbionts or at least chlorophyll fluorescence has a greater sensitivity to cooler temperatures than the coral host metabolism. Further work at this site could determine if symbiont identity influences the magnitude of cold-water bleaching as has been demonstrated by others (LaJeunesse et al., 2010). Declines in FV/ FM are commonly reported in symbiotic corals following cold stress (Saxby, Dennison & Hoegh-Guldberg, 2003; Kemp et al., 2011; Pontasch et al., 2017), less so changes in host physiology as in the current study and therefore suggests that the coral animal may tolerate longer periods of cold stress before mortality (but see Kemp et al. (2011) who report significantly reduced respiration rate of three species from Florida under an extreme cold stress of 12 °C). Reductions in FV/FM typically reflect dissipation of excess energy within the light harvesting antennae or photodamage to photosystem II. Changes in FV/FM can be detected soon after the onset of stress, e.g., after three hours at 20, 16, and 12  °C, Montipora digitata reduced FV/FM by 0.06, 0.15, and 0.22 respectively (Saxby, Dennison & Hoegh-Guldberg, 2003). Stylophora sp. from Lord Howe Island showed a significant 79.8% decline in FV/FM after 5 days’ exposure to 15 °C (Pontasch et al., 2017). Declining FV/FM was also noted following a 5 h acute cold stress in GoA species (supplementary information). Monitoring chlorophyll fluorescence therefore provides a fast, non-destructive, early indication of cold-water stress similar to how it has previously been used to indicate other environmental stressors (Cooper, Gilmour & Fabricius, 2009).

A temperature response curve created using published empirical data (Fine, Gildor & Genin, 2013; Bellworthy & Fine, 2017; Krueger et al., 2017) from corals collected at this site shows how FV/FM responds to temperature with ca. 2 weeks acclimation (Fig. 4). Highest FV/FM occurs between 27 and 28.5 °C (polynomial trend line, order 2) slightly above the maximum monthly mean for Eilat of 26.75 °C (Bellworthy & Fine, 2017), adding support to the idea that corals at this site live below their thermal optimum and are adapted to higher temperatures (Fine, Gildor & Genin, 2013; Krueger et al., 2017). In addition, Fig. 4 indicates that the lowest quantum yield occurs at the lowest experimental temperatures.

This study reports significant changes in other aspects of photochemistry not reported by previous studies (Table 2, Fig. 2). Reductions in maximal electron transport rate (rETRmax), saturation irradiance (IK), and photosynthetic efficiency (α) were noted for both species indicating inhibition of the photosynthetic apparatus under cold stress. Reductions in these parameters are often taken to infer high light stress and/ or high temperature stress (Warner, Fitt & Schmidt, 1996; Bischof, Hanelt & Wiencke, 2000; Fitt, Brown & Warner, 2001; Beer et al., 2006) or acclimation to low irradiance (Anthony & Hoegh-Guldberg, 2003; Cooper & Ulstrup, 2009; Ross et al., 2018). As corals in this study were acclimated to outdoor aquaria (with similar PAR level to their collection site) before the onset of cold stress, this is unlikely the cause here. Therefore, whilst the mechanism of damage to the photosynthetic apparatus under cold stress is potentially symbiont type specific (Pontasch et al., 2017) and different than under high light or heat stress, the results suggest that the physiological outcome is similar i.e., reduction in symbiont density and photosynthetic capacity. Cellular mechanisms common to warm and cold-water bleaching responses may include damage to photosystem II reaction centres (Wicks, Hill & Davy, 2010), changes in membrane fluidity (Tchernov et al., 2004; Kemp et al., 2011), or RUBISCO enzyme activity (Saxby, Dennison & Hoegh-Guldberg, 2003). Further work to describe cellular level thermal responses is likely to enhance scientific understanding.

Table 2 Statistical analyses of differences in physiological parameters between treatments within each experimental period (winter cold stress and summer warm period) for each species.

Results were obtained from linear mixed models where model residuals were normally distributed with equal variance between groups (fixed factor = treatment, random factor = genotype).

Parameter	Acropora eurystoma	Stylophora pistillata	
	Winter cold stress	Summer heat shock	Winter cold stress	Summer heat shock	
FV/ FM	m: 0.527 c: 0.527	F: 9.470 p: <0.0016	m: 0.188 c: 0.188	F: 1.971 p: >0.05	m: 0.382 c: 0.463	F: 6.046 p: <0.02	m: 0.014 c: 0.069	F: 0.167 †p: >0.05	
rETRmax	m: 0.781 c: 0.846	F: 39.435 p: <0.0001	m: 0.676 c: 0.676	F: 17.701 p: <0.0001	m: 0.667 c: 0.820	F: 31.456 p: <0.0001	m: 0.050 c: 0.638	F: 1.700 p: >0.05	
I k	m: 0.828 c: 0.828	F: 40.955 †p: <0.001	m: 0.214 c: 0.214	F: 2.309 †p: >0.05	m: 0.674 c: 0.918	F: 693509 p: <0.0001	m: 0.337 c: 0.647	F: 8.101 p: <0.01	
Alpha	m: 0.759 c: 0.759	F: 26.733 p: <0.0001	m: 0.081 c: 0.354	F: 1.150 p: >0.05	m: 0.351 c: 0.351	F: 4.594 p: <0.05	m: 0.789 c: 0.800	F: 33.437 p: <0.0001	
Chlorophyll / cm2	m: 0.256 c: 0.256	F: 4.136 p: <0.05	m: 0.144 c: 0.358	F: 1.908 p: >0.05	m: 0.388 c: 0.388	F: 5.077 p: <0.02	m: 0.169 c: 0.169	F: 61.789 p: >0.05	
Symbionts / cm2	m: 0.097 c: 0.316	F: 2.140 p: >0.05	m: 0.484 c: 0.484	F: 7.976 p: <0.001	m: 0.320 c: 0.320	F: 95.746 †p: <0.004	m: 0.042 c: 0.260	F: 0.485 p: >0.05	
Protein / cm2	m: 0.188 c: 0.253	F: 9.470 p: >0.05	m: 0.081 c: 0.085	F: 0.750 p: >0.05	m: 0.231 c: 0.231	F: 2.258 p: >0.05	m: 0.087 c: 0.087	F: 0.806 p: >0.05	
Respiration rate / cm2	m: 0.162 c: 0.162	F: 9.470 p: >0.053	m: 0.041 c: 0.263	F: 0.467 p: >0.05	m: 0.162 c: 0.162	F: 1.680 †p: >0.05	m: 0.118 c: 0.300	F: 1.427 †p: >0.05	
Photosyn thesis. rate/ cm2	m: 0.588 c: 0.600	F: 9.470 †p: <0.002	m: 0.206 c: 0.519	F: 3.640 p: >0.05	m: 0.832 c: 0.832	F: 39.723 p: <0.0001	m: 0.260 c: 0.364	F: 3.469 †p: >0.05	
Notes.

Where data did not meet assumptions of parametric tests, permanova models were performed as indicated by †.

Bold p values indicate statistical significance (α = 0.05). ’m’: Marginal R2 value, the proportion of variance explained by fixed factors alone. ’c’: Conditional R2 value, the proportion of variance explained by both fixed and random factors combined.

Protein concentration did not decrease after two weeks cold stress despite significant decreases in symbiont photochemical efficiency and net photosynthesis rates (Table 2, Fig. 3). A similar result was noted for Montipora digitata where Fv/ Fm decreased more than 50% from initial values after two weeks at 17.6 °C and yet did not display any corresponding decrease in protein concentration (Kavousi, Parkinson & Nakamura, 2016). Maintenance of total protein concentration and absence of mortality despite reduced energy production may be possible since metabolism, and therefore energy demand, decreases at lower temperatures (Schulte, Healy & Fangue, 2011; Schulte, 2015) as has been demonstrated in corals (e.g., (Coles & Jokiel, 1977; Haryanti & Hidaka, 2015)). Declining metabolism is not evident in this study. Alternatively, but not necessarily mutually exclusive, corals may have metabolised stored lipids (Rodrigues & Grottoli, 2007) or increased heterotrophic feeding (though corals were not explicitly provided with food, small zooplankton may enter the experimental aquaria through the 500 µm seawater filter). Plasticity in energy acquisition modes is known to be important in determining the outcome of thermal stress in corals (Hughes & Grottoli, 2013) not only in terms of mortality but also sub lethal processes such as growth and reproduction. Finally, increased heterotrophy during bleaching, as has been demonstrated under high temperature stress (Grottoli, Rodrigues & Palardy, 2006), may sustain coral animal functioning particularly if cold stress cooccurs with the winter spring plankton bloom.

The two studied species displayed similar responses to each other when experiencing either below or above ambient temperatures. This contrasts previous species-specific responses observed in the field. For example, at Malpelo Island in Colombia, the only species to bleach in the winter of 2008 was Porites lobata with 31.4% of colonies affected (Zapata, Jaramillo-González & Navas-Camacho, 2011). Even within the genus Acropora, some species are more cold-tolerant than others (Higuchi et al., 2015). Temperature tolerance of conspecifics appears to be site specific and a function of local thermal adaptation rather than taxonomy. In contrast to the aforementioned Colombian reefs, within the coldest regions of the Arabian Gulf Porites sp. dominates coral cover and has minimal bleaching whereas Acropora and Stylophora colonies have limited cover and suffer severe mortality following cold stress (Coles & Fadlallah, 1991). Testing other species from the diverse Red Sea may yet identify species specific responses though shallow water species from this site so far show a surprisingly consistent response to heat stress. Additionally, though not explicitly tested, based on prior knowledge (Byler et al., 2013) it is likely that all corals in this study hosted algal symbionts of the same genus. Since cold stress primarily affected the symbionts in this study, potentially the same coral species with a different symbiont may have a contrasting cold-stress response as demonstrated elsewhere (LaJeunesse et al., 2010). Correspondingly, the cold stress response of deeper (mesophotic) corals which host different symbiont genera at this study site, remains to be explored (Byler et al., 2013; Scucchia et al., 2020). All open questions raised here will likely be pertinent future research as both hot and cold thermal extremes increase in magnitude and frequency.

Figure 4 Thermal performance curve of maximum quantum yield of photosystem II (FV∕FM) of corals in Eilat.

Data are from published ex situ experiments including the current study (purple), Krueger et al., 2017 (yellow), Fine, Gildor & Genin, 2013 (green), and Bellworthy & Fine, 2017 (blue). Data points represent multiple species with thermal incubation times between 16 and 47 days (n = 149). The thick black line indicates the smoothed conditional mean with 99% confidence intervals (grey border). Dotted vertical lines indicate annual average minimum (Tmin) and maximum (Tmax) reef seawater temperatures in Eilat taken from the current paper and Bellworthy & Fine (2017) respectively.

Seasonal changes in coral physiology (Fitt et al., 2000) may sometimes be misinterpreted as stress symptoms, highlighting the importance of generating seasonal physiological baselines at different spatial scales (local, regional, and beyond). For example, in this study protein concentration, chlorophyll concentration, and net photosynthetic rate all appear reduced in the summer compared to wintertime for S. pistillata (Figs. 1 and 3). In A. eurystoma, net photosynthetic rate also appears depressed in the summer (Fig. 1). Nevertheless, there were few significant differences between treatment groups following the warm period indicating that exposure to winter cold stress does not impact corals’ response to above ambient summer temperatures. These results add further conceptual support to the Red Sea Reef Refuge Hypothesis of Fine, Gildor & Genin (2013). The evolutionary mechanism behind this hypothesis suggests that coral recruits resistant to high water temperatures established the present-day Red Sea population. Thus, these corals had little evolutionary experience to build cold water tolerance. Whilst GoA corals are rightly celebrated for presently living relatively far below their upper bleaching threshold (Fine, Gildor & Genin, 2013) and tolerate a relatively wide natural temperature range (ca. 21–27 °C), bleaching in this study has shown that their thermal plasticity is not boundless. In fact, GoA corals live relatively close to their lower bleaching temperature relative to other reefs (Table S1). Interestingly, Scleractinian species that show greatest mean summer mortality in the Florida reef tract have lowest mortality following a cold winter (Lirman et al., 2011). Here we suggest that the opposite trade off of high temperature tolerance with cold-water sensitivity may be present in GoA corals. We hypothesise that along the latitudinal and thermal gradient in the Red Sea, corals live closer to their upper thermal limits towards the south and the opposite latitudinal trend exists for cold temperature thresholds i.e., northernmost reefs are more likely to experience cold water bleaching.

The Red Sea is warming rapidly with the northern section warming faster than the global average. Some models predict increasing frequency and intensity of heat waves (Genevier et al., 2019) whilst others suggest the present warming trend will be counteracted due to natural climate oscillations shifting the Red Sea into a cooling phase in the upcoming decades (Krokos et al., 2019). Contradictions between available models add uncertainty to our ability to predict whether cold stress events will become problematic in the GoA. If an extreme cold weather scenario materializes, it will put the acclimation and longer-term adaptation potential of these corals to the test. Whilst corals of the GoA presently live relatively far from their upper thermal threshold (Fine, Gildor & Genin, 2013; Bellworthy & Fine, 2017; Krueger et al., 2017), they are in fact precariously close to their cold-water bleaching temperature.

Supplemental Information

Supplemental Information 1 Published records of cold water events causing coral bleaching either in the field or in corals obtained from the field for immediate experimentation ex situ

MinMM: minimum mean monthly temperature i.e., typical winter minimum. Gap: difference between MinMM and recorded bleaching temperature.

Click here for additional data file.

Supplemental Information 2 Additional short term acute cold stress experiment on Stylophora pistillata, and massive Dipsastraea favus, and Porites sp

Click here for additional data file.

Supplemental Information 3 Maximum quantum yield of photosystem II (FV/FM) in Dipsastraea favus sp., Porites sp., and Stylophora pistillata following exposure to acute cold stress

Bars show mean ± s.e. (n = 4 –5 corals per species). FV∕FM was measured on dark adapted corals at ambient 21° C, and again on the same corals following two hours at 19, 18, and 16°C. Lowercase letters indicate significant TukeyHSD post hoc differences in FV∕FM between experimental temperatures within a species.

Click here for additional data file.

We are grateful to Dror Komet for assistance with collection of the corals and maintenance of the Red Sea Simulator System. Many thanks to colleagues and three reviewers who reviewed and improved the manuscript prior to submission. In addition, thanks go to Roi Holzman and Tal Perevolotsky for guidance in completing the statistical analyses along with Ronen Liberman who supported the construction of graphical elements.

Additional Information and Declarations

Competing Interests

Author Contributions

Field Study Permissions

Data Availability

The authors declare there are no competing interests.

Jessica Bellworthy conceived and designed the experiments, performed the experiments, analyzed the data, prepared figures and/or tables, authored or reviewed drafts of the paper, and approved the final draft.

Maoz Fine conceived and designed the experiments, authored or reviewed drafts of the paper, and approved the final draft.

The following information was supplied relating to field study approvals (i.e., approving body and any reference numbers):

Sampling was conducted under permit no. 2018/41868 from the Israel Nature Parks Authority.

The following information was supplied regarding data availability:

All raw data and R code are available at GitHub:

https://github.com/JessicaBellworthy/Bellworthy_Corals_Eilat_ColdStress.

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
