# Peer review of "Warming resistant corals from the Gulf of Aqaba live close to their cold-water bleaching threshold"

_PeerJ, doi:10.7717/peerj.11100_

## Round 0.1 · original submission · Minor Revisions

All three reviewers agreed this was a solid, well-written study although they made several suggestions that you should give serious attention to (and also reviewer 3 attached a figure for your consideration). I have also made a few notes about the clarity of the introduction for non-specialists a comment about the figures.

Lines 59-60: Can you clarify why lower temperatures thought to be 18, 16 or 14C, but cold period with temps below 11C resulted in only 26% mortality (compared to 16C causing 91% mortality).

Lines 73-77: Can you clarify why selection in the southern Red Sea has caused resilient coral populations in the northern Red Sea.

Map of coral distribution in larger study region would be nice.

What about shallow vs. deeper water corals - how much do we know about their tolerance to cold, relative to average sea water temp rather than latitude?

Intro could use some clarity of details.

A bit jargony - can you look for opportunities to make it more digestible for an audience without a strong background in coral ecology and physiology?

95-101: Not sure you need to explicitly state the null hypotheses separate from the motivating questions (but, if you do, look to restructure the flow of that paragraph, which is awkward).

108-111 do not present the most useful metrics. minimum monthly mean temperature perhaps not as meaningful as avg. minimum temperature of coldest quarter (or similar).

Should there also be a block random effect for each aquarium?

Fig. 1 - label columns with species and add a legend for color meaning (more clear description is color reflects winter temperature treatments, which varied; summer treatments were the same for all)

Reviewer 1 ·

Basic reporting

The language of the manuscript is clear and professional throughout. The introduction covers the background well, and the discussion aids in interpreting the results in relation to the knowledge gap outlined in the intro. The discussion around contrasts in physiological responses between the symbiont community and host was particularly interesting.

Experimental design

This manuscript presents original research within the scope of the journal. The research questions and resulting hypotheses are clearly stated and flow logically from the background information presented in the introduction.

The experimental approach is appropriate to address the research questions. However, when the authors outline their two experimental phases (lines 148-167), it is clear that less than 3 months occurred between the “winter” and “summer” phase. What was the justification for this period of time between phases? Is this based on the average duration between minimum and maximum temperatures in the region, or was there some other consideration made? I fully support the authors in the longer hold at ambient between phases but I am wondering if 3 months was appropriate or if it should have been longer? Although I expect that a longer hold would not have significantly changed the outcomes of the summer phase.

Validity of the findings

The results are presented, interpreted and then discussed in an appropriate manner. The conclusions are well-supported by the data.

However, I was surprised that in some cases, there was no effect of genotype, based on some models producing equal marginal and conditional R2 values. The corals were collected from a nursery (lines 105-106), so I am wondering if they have been established previously as genetic individuals or whether it could be possible that some of the genotypes were actually clones? It is also interesting that in some of the cases where there is a difference between the marginal and conditional R2 values, genotype effects account for as much as 30% of the variation in the data (specifically, A. eurystoma, summer heat shock, photosynthesis).

Additional comments

Overall, the authors are presenting a very interesting study, highlighting a little-studied issue in which corals that are high-heat adapted may or may not be cold-adapted, too. This is an important issue to consider given the projected increase in severe cold-weather events under future climate change.

Reviewer 2 ·

Basic reporting

Well-written and succinct.

Experimental design

This is a simple, elegant experiment to assess the lower physiological thermal bounds of corals in the Gulf of Aqaba and any potential links between low- and high-temperature resilience. It builds on previous work and is a nice contribution to the literature.

Validity of the findings

The author's evaluation of their data is appropriate, see general comments.

Additional comments

The authors have conducted a small but well-conceived experiment on cold-water stress on coral physiology. Overall I found the experiment conducted and analysed well and a good contribution to the rather limited literature of low-temperature coral stress.

My primary criticism is the description of the summer high-temperature exposure as "acute thermal shock". The link between cold stress and subsequent high-temperature stress seems flawed in that the summer samples do not appear to be stressed; the quantum yields are high and cell densities seem largely unchanged except for the 18.6C treatment in the summer phase in Acropora. The other parameters seem to be what would be expected of corals in the summer. Is +3 degrees above the monthly mean the same as +3 over annual mean (i.e. is June the warmest month)? Based on Fig. 4 the corals are still several degrees from the summer maximum. In short: are the authors convinced that their 30C treatment was a stressful event? Was the temperature high enough and sustained for enough time? If not, then the link between low- and high-temperature treatment need to be re-evaluated. It's certainly true that the cold treatment did not make them especially sensitive, however. In any case I don’t think “acute thermal shock” is the correct term given these physiological data.

Line 56: I disagree that the mechanisms of warm and cold thermal stress are the same. High-temperature stress alters protein folding processes, while low-temperature stress can slow/inhibit multiple (or all) cellular processes. Membrane fluidity is likely a factor in both.

The paper would also benefit from a very brief hypothesis in the discussion, clearly marked as speculation, as to what cellular mechanisms might link low and high temperature responses. I understand that this is an ecophysiology paper, but a sentence or two on what processes are shared between the two (even if it has to be drawn from non-coral literature, e.g. plant) would strengthen the manuscript.

Figure 3 B & D: The increase in R at low temperatures is odd. Is this due to highly variable measurements?

Figure 4: Are these temperatures experimental or from the field, that is, is 34C experienced by corals in GoA? If not, some vertical lines in the figure to denote the typical thermal range of the corals used in these experiments would make the figure more clear. Different shapes of experimental vs. field points might also improve the figure.

282 Host biomass (as measured by protein concentration) was maintained.
356 tested

Reviewer 3 ·

Basic reporting

This manuscript addresses cold-stress on symbiotic corals living in the Gulf of Aqaba. The researchers found that the symbiont density, chlorophyll content, and photosynthetic rate were significantly lower at temperatures that were reduced 1, 3 oC compared to ambient wintertime temperatures for both S. pistillata and A. eurystoma, implying that these corals are living close to their cold-water bleaching limits. The Fv/Fm, ETR data show that the algae is compromised at 1oC, and definitely at 3oC lower than ambient in the winter.

That said, there are a few things that seem a bit odd in this study:

1. The symbionts densities seem a bit low. I remember them being about 1 x 106 /cm2 for S. pistillata (e.g. Muscatine 1980’s). Perhaps because the corals were collected from “a coral nursery” adjacent to the Eilat lab? Perhaps because they were collected from deeper water?

2. Line 117: the authors say that the symbiont was Symbiodinium and reference a 2013 paper. Todd LaJuenesse has recently reorganized the Symbiodiniaceae (Current Biology 28:2570, 2018), with Symbioinium microadriaticum probably inhabiting S. pistillata, and Cladocopium sp. living with the Acropora. Unless the authors have more recent news, I would use this.

3. Line 365: I do not see the “protein concentration, chlorophyll concentration, and net photosynthetic rate all appear reduced in the summer for S. pistillata”. Do you mean compared to wintertime data?

4. Line 377: “corals live relatively close to the lower bleaching temperature”. I think this is true for most northerly coral populations (out of the tropics). Many of the corals on the Florida Reef Tract might actually die before they actually bleach. See third graph next page!

5. Line 390-3: There is more variation around the mean temperatures during global warming. This means that temperatures potentially go higher in summer, lower in winter. Let’s see…could go either way?

This manuscript seems a bit pedestrian, but certainly publishable!

Experimental design

This manuscript addresses cold-stress on symbiotic corals living in the Gulf of Aqaba. The researchers found that the symbiont density, chlorophyll content, and photosynthetic rate were significantly lower at temperatures that were reduced 1, 3 oC compared to ambient wintertime temperatures for both S. pistillata and A. eurystoma, implying that these corals are living close to their cold-water bleaching limits. The Fv/Fm, ETR data show that the algae is compromised at 1oC, and definitely at 3oC lower than ambient in the winter.

That said, there are a few things that seem a bit odd in this study:

1. The symbionts densities seem a bit low. I remember them being about 1 x 106 /cm2 for S. pistillata (e.g. Muscatine 1980’s). Perhaps because the corals were collected from “a coral nursery” adjacent to the Eilat lab? Perhaps because they were collected from deeper water?

2. Line 117: the authors say that the symbiont was Symbiodinium and reference a 2013 paper. Todd LaJuenesse has recently reorganized the Symbiodiniaceae (Current Biology 28:2570, 2018), with Symbioinium microadriaticum probably inhabiting S. pistillata, and Cladocopium sp. living with the Acropora. Unless the authors have more recent news, I would use this.

3. Line 365: I do not see the “protein concentration, chlorophyll concentration, and net photosynthetic rate all appear reduced in the summer for S. pistillata”. Do you mean compared to wintertime data?

4. Line 377: “corals live relatively close to the lower bleaching temperature”. I think this is true for most northerly coral populations (out of the tropics). Many of the corals on the Florida Reef Tract might actually die before they actually bleach. See third graph next page!

5. Line 390-3: There is more variation around the mean temperatures during global warming. This means that temperatures potentially go higher in summer, lower in winter. Let’s see…could go either way?

This manuscript seems a bit pedestrian, but certainly publishable!

Validity of the findings

This manuscript addresses cold-stress on symbiotic corals living in the Gulf of Aqaba. The researchers found that the symbiont density, chlorophyll content, and photosynthetic rate were significantly lower at temperatures that were reduced 1, 3 oC compared to ambient wintertime temperatures for both S. pistillata and A. eurystoma, implying that these corals are living close to their cold-water bleaching limits. The Fv/Fm, ETR data show that the algae is compromised at 1oC, and definitely at 3oC lower than ambient in the winter.

That said, there are a few things that seem a bit odd in this study:

1. The symbionts densities seem a bit low. I remember them being about 1 x 106 /cm2 for S. pistillata (e.g. Muscatine 1980’s). Perhaps because the corals were collected from “a coral nursery” adjacent to the Eilat lab? Perhaps because they were collected from deeper water?

2. Line 117: the authors say that the symbiont was Symbiodinium and reference a 2013 paper. Todd LaJuenesse has recently reorganized the Symbiodiniaceae (Current Biology 28:2570, 2018), with Symbioinium microadriaticum probably inhabiting S. pistillata, and Cladocopium sp. living with the Acropora. Unless the authors have more recent news, I would use this.

3. Line 365: I do not see the “protein concentration, chlorophyll concentration, and net photosynthetic rate all appear reduced in the summer for S. pistillata”. Do you mean compared to wintertime data?

4. Line 377: “corals live relatively close to the lower bleaching temperature”. I think this is true for most northerly coral populations (out of the tropics). Many of the corals on the Florida Reef Tract might actually die before they actually bleach. See third graph next page!

5. Line 390-3: There is more variation around the mean temperatures during global warming. This means that temperatures potentially go higher in summer, lower in winter. Let’s see…could go either way?

This manuscript seems a bit pedestrian, but certainly publishable!

Additional comments

This manuscript addresses cold-stress on symbiotic corals living in the Gulf of Aqaba. The researchers found that the symbiont density, chlorophyll content, and photosynthetic rate were significantly lower at temperatures that were reduced 1, 3 oC compared to ambient wintertime temperatures for both S. pistillata and A. eurystoma, implying that these corals are living close to their cold-water bleaching limits. The Fv/Fm, ETR data show that the algae is compromised at 1oC, and definitely at 3oC lower than ambient in the winter.

That said, there are a few things that seem a bit odd in this study:

1. The symbionts densities seem a bit low. I remember them being about 1 x 106 /cm2 for S. pistillata (e.g. Muscatine 1980’s). Perhaps because the corals were collected from “a coral nursery” adjacent to the Eilat lab? Perhaps because they were collected from deeper water?

2. Line 117: the authors say that the symbiont was Symbiodinium and reference a 2013 paper. Todd LaJuenesse has recently reorganized the Symbiodiniaceae (Current Biology 28:2570, 2018), with Symbioinium microadriaticum probably inhabiting S. pistillata, and Cladocopium sp. living with the Acropora. Unless the authors have more recent news, I would use this.

3. Line 365: I do not see the “protein concentration, chlorophyll concentration, and net photosynthetic rate all appear reduced in the summer for S. pistillata”. Do you mean compared to wintertime data?

4. Line 377: “corals live relatively close to the lower bleaching temperature”. I think this is true for most northerly coral populations (out of the tropics). Many of the corals on the Florida Reef Tract might actually die before they actually bleach. See third graph next page!

5. Line 390-3: There is more variation around the mean temperatures during global warming. This means that temperatures potentially go higher in summer, lower in winter. Let’s see…could go either way?

This manuscript seems a bit pedestrian, but certainly publishable!

Annotated reviews are not available for download in order to protect the identity of reviewers who chose to remain anonymous.

---

## Round 0.2 · accepted · Accept

Thanks for your careful consideration of the comments. This study makes an important contribution to our understanding of temperature stress in response to increasingly variable environments. Congratulations!!

Reviewer 1 ·

Basic reporting

The authors have adequately addressed concerns raised by all three reviewers.

Experimental design

Appropriate

Validity of the findings

Appropriate

Additional comments

Minor edit on line 275.

Reviewer 2 ·

Basic reporting

Appropriate.

Experimental design

Appropriate.

Validity of the findings

Appropriate interpretations.

Additional comments

The authors have made the requested changes and I'm satisfied with the manuscript.

Reviewer 3 ·

Basic reporting

I have read the rebuttal letter, with the 3 reviewer comments/authors comments.

Experimental design

I have read the tracked changes manuscript. Minor changes made by authors.

Validity of the findings

It is my opinion that the authors addressed most of the comments.

Additional comments

Nothing more to add!